# Variation in the Drought Tolerance of Tropical Understory Plant Communities across an Extreme Elevation and Precipitation Gradient

**DOI:** 10.3390/plants12162957

**Published:** 2023-08-16

**Authors:** Catherine H. Bravo-Avila, Kenneth J. Feeley

**Affiliations:** 1Department of Biology, University of Miami, Coral Gables, FL 33146, USA; 2Fairchild Tropical Botanical Garden, Coral Gables, FL 33156, USA

**Keywords:** Andean forests, drought tolerance, elevation gradient, solute leakage, tropical forest, understory plants

## Abstract

Little is known about how differences in water availability within the “super humid” tropics can influence the physiology of understory plant species and the composition of understory plant communities. We investigated the variation in the physiological drought tolerances of hundreds of understory plants in dozens of plant communities across an extreme elevation and precipitation gradient. Specifically, we established 58 understory plots along a gradient of 400–3600 m asl elevation and 1000–6000 mm yr^−1^ rainfall in and around Manu National Park in southeastern Peru. Within the plots, we sampled all understory woody plants and measured three metrics of physiological leaf drought tolerance—turgor loss point (TLP), cuticular conductance (G_min_), and solute leakage (SL)—and assessed how the community-level means of these three traits related to the mean annual precipitation (MAP) and elevation (along the study gradient, the temperature decreases linearly, and the vapor pressure deficit increases monotonically with elevation). We did not find any correlations between the three metrics of leaf drought tolerance, suggesting that they represent independent strategies for coping with a low water availability. Despite being widely used metrics of leaf drought tolerance, neither the TLP nor G_min_ showed any significant relationships with elevation or the MAP. In contrast, SL, which has only recently been developed for use in ecological field studies, increased significantly at higher precipitations and at lower elevations (i.e., plants in colder and drier habitats have a lower average SL, indicating greater drought tolerances). Our results illustrate that differences in water availability may affect the physiology of tropical montane plants and thus play a strong role in structuring plant communities even in the super humid tropics. Our results also highlight the potential for SL assays to be efficient and effective tools for measuring drought tolerances in the field.

## 1. Introduction

Elevational and environmental gradients can be powerful tools for investigating the importance of habitat filtering and community assembly processes. Indeed, the potential value of environmental gradients in demonstrating how different factors drive patterns of plant diversity and vegetation structure has long been recognized [1,2], and previous studies of environmental gradients have provided evidence that many plant species have different distributions based on their tolerances to climate, topography, and soil characteristics [3,4,5,6].

Studies of elevational gradients can also provide important insights into how tropical species, communities, and ecosystems are responding to ongoing anthropogenic climate change [7,8,9,10,11,12]. Elevational gradients provide particular insight into the influence of temperature because air and soil temperatures typically decrease linearly with elevation [12,13]. In the Amazon–Andes region, elevational gradients have been used to assess the effects of temperature on ecosystem productivity and community structure [8,14,15,16,17,18] and to predict how species and communities will be affected by climate change [19,20].

The importance of changes in water availability through time for driving shifts in species’ distributions is widely acknowledged in many temperate ecosystems [21,22,23,24,25,26]. Some previous studies have also shown that changes in precipitation are driving directional changes in the community composition of tropical forests [27,28,29], but other studies have shown little or no relationship between changes in species composition and changes in precipitation [11], and it has even been suggested that observed changes in composition in relation to drought tolerances may be driven primarily by changes in temperature and the inherent relationships between species’ heat and drought tolerances (especially when measured based on geographic occurrence locations) [30]. The hypothesized drivers of these heterogeneous responses are complex changes in the climatic water balance (e.g., cloud cover immersion, climatic water deficit, and seasonal precipitation), emphasizing the potential importance of altered water availability [31]. Given that climate change is altering tropical precipitation regimes [32,33], this incomplete knowledge about how important water availability is for determining species’ current and future distributions greatly limits our ability to predict the fate of these tropical communities and ecosystems.

Rainfall is commonly used as a proxy for water availability. Soil water deficit can occur when evapotranspiration exceeds the seasonal or annual precipitation; as a result, the soil moisture is depleted, and many water-dependent physiological processes in plants are negatively impacted [34]. The air temperature can also drive changes in the atmospheric water availability. When the air temperature increases, the gradient of leaf-to-air vapor pressure deficit (VPD) increases, increasing the rate of water loss via evapotranspiration. Many plants respond to an increased VPD through stomatal closure, with consequent negative effects on the photosynthetic rates and plant growth [35]. Indeed, the VPD, which depends on both humidity and temperature, has been increasingly identified as an important measure of water availability, and increases in the VPD are considered a major contributor to drought-induced plant mortality [36].

Since plant water availability is a complex phenomenon involving the interplay between multiple climate variables, soil and environmental factors, and intrinsic species traits related to water uptake and use, the characterization of water availability is nearly impossible for diverse species assemblages spread across large environmental gradients. Water availability is generally believed to decrease at high elevations in the tropics due to reduced rainfall, increased rainfall seasonality, and thinner soil organic material [37,38]; however, changes in water availability across elevations are not necessarily monotonic or universal. Lowland rainforests can have heavy rains and high temperatures, but these climate variables are modulated by the soil water capacity [39]. In the sub-montane forests (above lowland rainforests), there is generally higher precipitation, but the VPD remains high and there may be increased seasonality at mid-elevation levels [13]. Cloud forests, which occur above the sub-montane forests, can have a high water availability due to high relative humidity and regular cloud and fog immersion (although there can be seasonal periods with reduced fog, leading to high irradiance and increased VPD) [13]. As a consequence of regular cloud immersion and a high water availability, cloud forests generally have thicker soil organic layers, abundant mosses, and high densities of epiphytes [40,41,42]. Above the cloud forests, the habitat transitions to high-elevation grasslands (in the southern Peruvian Andes, these grasslands are referred to as “puna”) where the climate is drier and colder, the precipitation is more seasonal, and the soils tend to have a thinner organic layer [43]. Embedded in these high-elevation grasslands, there can be stands of shrubs and small patches of forests where the local climate is more like that of the cloud forests [44]. These changes in water availability across elevations should lead to strong habitat filtering of plant species based on their specific habitat requirements and differences in drought tolerances.

As the water availability decreases, plant hydraulic conductance may decrease through different biophysical and physiological mechanisms to prevent embolism or desiccation [45]. In this study, we used the turgor loss point, cuticular conductance, and solute leakage as tools to characterize the relative leaf drought tolerances of individual understory plants and understory plant communities. The turgor loss point (TLP), which is the leaf water potential at which cell turgor is lost and leaf wilting occurs, provides physiological information pertaining to the cell wall integrity, stomatal closure, and, more generally, the extent to which the plants can maintain metabolism as the water availability decreases due to soil drying [46]. As such, plant drought tolerances should be negatively correlated with the TLP (i.e., plants that can withstand greater negative water potentials before wilting are considered to be more drought tolerant than ecologically similar plants that wilt at less negative water potentials) [47].

Another mechanism that can promote drought tolerance is avoiding desiccation and maintaining a high relative water content, thereby resisting the cell damage that can be caused by dehydration. Under dry conditions, plants may close their stomata, relying on their cuticle to prevent further water loss. Cuticular conductance (G_min_) is the conductance to vapor diffusion across the leaf epidermis (i.e., through the cuticle and any leaky stomata) once the stomata are closed [48]. Plants with a low G_min_ are generally considered to be more drought tolerant than plants with a high G_min_ (although previous studies have shown high interspecific variability in G_min_ that was not associated with differences in rainfall [49,50]).

Responses at the cellular level may also prevent damage to plants during droughts [51,52,53]. The measurement of the solute leakage (SL) from plant tissues is a proven method to detect cytorrhysis (damage of cell walls) under laboratory conditions causing desiccation. As such, measurements of the SL from leaves exposed to standardized desiccating conditions is a potential metric of relative drought tolerances. Indeed, the SL has previously been shown to be an accurate indicator of the drought and salt tolerances of crop plants [54,55,56,57,58]. However, SL has rarely been used in ecological field studies. Fadrique et al. (2022) worked along the same elevational gradient in Peru and found that the SL of bamboo plants decreased at higher elevations and under wetter conditions [59]. Zuleta et al. (2022) analyzed the intra and interspecific differences of the SL in trees growing in different topographic habitats in the lowland Amazon rainforest and found that trees growing in wetter microhabitats had a higher SL than the trees growing on the drier ridge tops [60].

In this study, we measured the leaf TLP, G_min_, and SL in hundreds of individual understory saplings and shrubs and assessed the community-level metrics of leaf drought tolerance in approximately 60 plots located along a 3000+ m elevation and a 5000 mm yr^−1^ rainfall gradient from lowland rainforests to highland gallery forests and high-elevation shrubs. We tested the relationships of these community drought tolerance metrics with elevation (a proxy for temperature and the VPD; the VPD decreases at higher elevations due to lower air temperatures) and the mean annual precipitation. Specifically, we addressed the two following questions and corresponding a priori hypotheses: (1) Are the three metrics of drought tolerance coordinated or independent of each other? We hypothesized that the TLP, G_min_, and SL would all be correlated at the individual plant and plot levels due to a coupling of leaf-level mechanisms to resist and tolerate drought [61]. (2) How is drought tolerance related to rainfall and elevation across the gradient? We hypothesized that since the adaptation and/or acclimation to drought can incur significant costs, drought-tolerant species would be relatively infrequent in sites with high rainfall, and plant communities at lower elevations with hotter climates would have greater average drought tolerances than plants at higher elevations. Increasing our understanding of the roles that water availability and drought tolerance play in structuring understory plant communities will improve our ability to predict the responses of tropical ecosystems to environmental changes.

## 2. Methods

We conducted our study of understory plants and plant communities along a steep elevational gradient in Manu National Park in southeastern Peru (Cusco and Madre de Dios regions; Figure 1). The study area ranged from 400 to 3600 m asl and included forests along the Trocha Union ridge in the Kosñipata Valley (900 to 3600 m asl) and around the Cocha Cashu Biological Station (400 m asl; Table 1). Most of the study region is characterized as being “super humid”; the relative orientation of prevailing winds and topography create a zone of frequent cloud immersion when cold Andean winds collide with warm, moist air from the Amazonian lowlands [62]. Mean annual temperature (MAT) ranges from ~23 °C at 400 m asl to <9 °C at 3400 m asl (Table 1), with a lapse rate for the understory of 0.53 °C per 100 m elevation [13]. Mean annual precipitation (MAP) has been measured at a series of meteorological stations along the transect and ranges from ~2000 mm yr^−1^ at low elevations to 5000 mm yr^−1^ at middle elevations and <1000 mm yr^−1^ at the very highest elevations. The MAP at each plot location was estimated based on the observed relationship of precipitation vs. elevation in nearby weather stations (adjusted R^2^ = 0.99) and interpolating to the plot locations based on their elevations as measured with a handheld GPS (Table 1) [13]. Mean VPD in the understory increases monotonically with elevation along the study gradient from 0.01 in the lowlands to 0.12 kPa in the highlands [13].

### 2.1. Vegetation Sampling

We established 58 understory plots of 5 × 5 m (25 m^2^) each. Approximately four plots were installed every 250 m in elevation between 900 and 3600 m asl. Four additional plots were installed at approximately 400 m asl in the lowland rainforest near the Cocha Cashu Biological Station (Figure 1), but because of logistical difficulties, these lowland plots were only used to measure SL (see below).

We defined the understory to include all woody shrubs and saplings with basal diameters ≥ 1 cm at ground level and at heights between 0.5 and 5 m (plant height was measured as the vertical distance from ground to the top of the plant canopy; in some cases, stem length was >5 m if the plant was leaning). For every individual, we measured basal diameter, total height, and diameter at breast height (dbh) when height was >1.3 m. Most individuals (90%) were identified to family level in the field, and some to genus level. Unidentified individuals were assigned a temporary name in the field. Subsequently, the unidentified individuals were compared with specimens in regional botanical collections and determined to species level or vouchered morphospecies. Vouchers were compared across all plots and taxonomy was standardized. Since all analyses were conducted at the plot level using the averages of individual-level measurements, the high frequency of morphospecies does not affect the results.

One terminal branch was collected from each individual understory plant in each plot and placed in black plastic bags along with wet paper towels inside, and stem ends were submerged in water in plastic bags to avoid desiccation during transport to temporary field laboratories established near the top and bottom of the main elevation gradient at 3500 m asl and at 1300 m asl, respectively, and at Cocha Cashu Biological Station at 400 m asl. Once in the “laboratory”, branches were placed in buckets, stems were recut under water, and leaves were allowed to rehydrate covered with plastic. For each individual, all traits were measured on mature leaves from the same branch. Leaves were only included if they looked healthy and were not discolored; any leaves that were noticeably affected by pathogens were excluded.

Turgor loss point (TLP). Before measurements, branches were allowed to rehydrate for 10 h. After rehydration, three mature leaves were collected from each branch and allowed to slowly desiccate on a table over a period of 6–8 h between 02:00 and 10:00 a.m. local time. During this desiccation period, leaf mass (g) and leaf water potential (*Ψ* _leaf_, MPa) were measured periodically. Leaf mass was measured with an analytical balance (±0.002 g, Ohaus, Parsippany, NJ, USA, Scout SPX223) and *Ψ* _leaf_ was measured using a Scholander-type pressure chamber (model 1000, PMS Instruments, Corvallis, OR, USA). Plotting the inverse of leaf water potential against relative water content allows for the determination of the turgor loss point (TLP) as the point of transition between linear and nonlinear portions of the pressure–volume (PV) curve. A lower TLP (i.e., more negative water potentials) indicates a higher resistance to leaf wilting [63]. We were not able to construct PV curves or determine TLP for all individuals or species because of short petioles, sap presence in the leaves, or logistical difficulties.

Cuticular conductance (G_min_). To measure the leaf G_min_, we harvested three fully expanded leaves from each individual’s sample branch after ≥4 h rehydration. We cut the leaf petioles and then sealed the petiole ends with paraffin. Leaves were dried at night using LED lanterns (PPFD 1.73–4.6 µmol m^−2^s^−1^) on a table for 1 h in order to induce stomatal closure. The leaves were then massed six times at intervals of 30 min. Cuticular conductance was calculated as the rate of water loss (g) over time [48]. At each weighing, we measured the air temperature (°C) and relative humidity (%) with a handheld sensor (Kestrel 2500 NV, Nielsen-Kellerman, Boothwyn, PA, USA) to calculate the saturation vapor pressure (VP_sat_, kPa). Following standard protocols, the value of G_min_ (mmol m^−2^ s^−1^) was calculated as the water loss rate divided by the VPD (mol mol^−1^) and twice the total leaf area as measured using scanned leaf images (cm^2^) [64]. A lower G_min_ indicates a high drought tolerance since a lower cuticular conductance enables the maintenance of hydration based on stored water.

Solute leakage (SL). To measure solute leakage (SL), we harvested five mature leaves from the sample branches of each individual after ≥4 h rehydration. We lightly cleaned each leaf to remove surface debris and cut three pieces from the apical, middle, and bottom areas of the leaf with a 1.27 cm diameter hole punch. We submerged the leaf pieces in 20 mL of 50% Polyethylene glycol 3350 (PEG), which is a hypertonic solution with an osmotic potential of approximately −8.4 MPa, for 12 h [65]. After the PEG treatment, the leaf pieces were rinsed three times with distilled water (2 µS) to remove any PEG residue. We then transferred the treated leaf pieces to 50 mL falcon tubes with distilled water and measured the initial electrolyte conductivity (C_0_) using an Oakton CON 6+ Meter conductivity meter. After 12 h in the distilled water, we took a third conductivity measurement (C_1_). The leaf pieces and water were then boiled in the sealed plastic tubes for 20 min to rupture any remaining intact cells. Twelve hours after boiling, we took a second conductivity measurement (C_2_). We calculated the relative solute leakage (%) as (C_1_ − C_0_)/(C_2_ − C_0_) × 100 [59,60]. Higher SL values indicate greater cytorrhysis when exposed to desiccating conditions and thus a lower relative leaf drought tolerance. This method assumes that solute leakage is the result of membrane breakage due to water stress within cells, that leaked ions are representative of initial endogenous concentrations in leaf samples before PEG treatment [66], and that differences in leaf nutrition do not influence the concentration of leaked ions [65]. In addition, it assumes that variation in solute leakage is not caused by variation in the capacity of water transport by aquaporins and ion transport by channel and carrier proteins embedded in the plasma membrane. Given that SL is related to several key hydraulic and biochemical parameters, including stomatal resistance and tissue osmotic potential [67], water use efficiency [68], and tissue nitrogen levels [69], we find that this is a valuable and accessible field tool for assessing relative drought tolerance across many individuals and species.

### 2.2. Statistical Analysis

We used three leaves per individual to calculate average TLP and G_min_ and five leaves per individual to obtain SL. The values for all individuals in each plot were then averaged to obtain community-weighted mean (CWM) trait values. We measured SL on each individual plant in each plot. In the case of TLP and G_min_, we were not able to sample every individual, and we therefore calculated the CWM trait values as the mean of the species-level average trait values weighted by the number of stems per species. Species-level averages were always based on measurements of individuals within each focal plot; in other words, drought tolerance measurements were never generalized between plots. Using plot-specific measurements accounts for any intraspecific variation due to local adaptation or acclimation and eliminates the need for taxonomic standardization across plots. We conducted Pearson correlation analyses between the traits to determine possible relationships between the different metrics of physiological drought tolerance. Linear regression analyses were used to assess the relationships between the plots’ CWMs of the SL, TLP, and G_min_ with the elevation and MAP. All analyses were conducted in R [70].

## 3. Results

### 3.1. Floristic composition

The 58 sample plots contained a total of 834 understory plants with 363 (morpho)species, representing 89 genera and 50 families (Table 1). The most abundant families in the plots were *Melastomataceae* (170 individuals), *Rubiaceae* (155 individuals), and *Chloranthaceae* (55 individuals). The most abundant genera were *Miconia* (165 individuals), *Hedyosmum* (55 individuals), and *Palicourea* (45 individuals). Of the 834 individuals, 501 were saplings and 333 were shrubs (Appendix A). On average, there were 14 individuals and 10 species per plot, but the species richness and stem density varied greatly across elevations and between habitats. The plots at mid-elevation levels in the sub-montane forest habitat had the greatest density of individuals and the highest number of species (Table 1; Figure 2).

### 3.2. Drought Tolerance Metrics

The values of the SL, G_min_, and TLP for each individual plant are reported in Appendix A (Appendix A also includes several other metrics derived from the pressure–volume curves but are not included in the analyses reported here). The community-weighted mean values of the SL, G_min,_ and TLP for each plot are reported in Table 2. The plot mean TLP varied from −0.36 MPa to −1.42 MPa (lower values indicate a higher drought tolerance) (Table 2). Due to logistical constraints and the difficulties of measuring the TLP in extremely remote locations lacking infrastructure, the TLP was only measured in 28 plots from middle and high elevations (937–3604 m asl; i.e., only in sub-montane forest, cloud forest, and gallery forest habitats). The mean G_min_ varied between plots from 4.18 to 99.11 mmol m^−2^ s^−1^ (lower values indicate a higher drought tolerance) (Table 2). As with the TLP, the G_min_ was only measured in middle and high elevations (1189–3604 m asl). The mean SL varied markedly among plots from 13% to 81% (Table 2). The SL was measured in all plots (400–3615 m asl).

There were no significant correlations between the three drought tolerance metrics at either the individual plant level (there was a marginally significant correlation at the individual level between the SL and G_min_, *p* = 0.06) or at the plot level (i.e., CWMs) (Table 3; Figure 3).

There were no significant relationships between the TLP or G_min_ with either elevation or the mean annual precipitation (Figure 4). In contrast, the solute leakage (SL) was significantly related to elevation (SL ~ 65.94 − 0.0067 × Elevation (in m); *p* < 0.00005; adjusted R^2^ = 0.23) and especially the MAP (SL ~ 27.23 + 0.0058 × MAP (in mm); *p* < 0.00005; adjusted R^2^ = 0.50), such that the SL was the lowest (indicating a greater drought tolerance) at high elevations and in areas with lower precipitation (Figure 4). When both the elevation and MAP were included as explanatory variables, only the MAP was significant. The significant relationships between the SL and precipitation remained even if we limited the analysis to the subset of plots where we also had information on the TLP and G_min_ (SL ~ 25.03 + 0.0062 × MAP; *p* < 0.00005; adjusted R^2^ = 0.55).

## 4. Discussion

Little is known about whether climatic variation within the super humid tropics can cause variation in the drought tolerance and community assembly of plant species. We assessed the changes in the relative drought tolerance of nearly 60 understory woody plant communities along a 3000+ m elevational gradient in the “super humid” tropics by characterizing the relationships of three different metrics of physiological leaf drought tolerance with precipitation and elevation. In total, we performed at least one measurement of drought tolerance on almost 800 individual understory plants and measured all three measures of drought tolerance on almost 100 individual plants (Appendix A). Using these data, we tested if the three metrics of drought tolerance coordinated with each other, and tested if the three drought tolerance metrics are related to rainfall and elevation across the gradient.

### 4.1. Are the Three Metrics of Drought Tolerance Coordinated or Independent of Each Other?

Our a priori hypothesis was that the TLP, G_min_, and SL would all be correlated at the individual and plot levels due to a coupling of leaf-level mechanisms to resist and tolerate drought.

Contrary to this hypothesis, we did not find covariation between our three metrics of drought tolerance at either the plot level or the individual level (Table 3; Figure 3). This suggests that the plants in this system may have developed different independent mechanisms of tolerating and avoiding drought stress. This is contrary to previous studies in other systems that found coordination among the stomatal, hydraulic, and leaf traits associated with drought tolerance [60,71,72]. Additional research is required to understand the interactions and tradeoffs in the different drought tolerances and avoidance mechanisms employed by tropical montane plant species and the implications of these mechanisms for plant performance, as well as the potential effects of other abiotic and biotic factors (e.g., edaphic factors and competition, respectively).

### 4.2. How Are Drought Tolerance Metrics Related to Rainfall and Elevation across the Gradient?

Our a priori hypothesis was that drought-tolerant species would be relatively infrequent in sites with high rainfall and that plant communities at lower elevations with hotter climates would have greater average drought tolerances than plants at higher elevations.

The turgor loss point (TLP) values that we measured in the understory plants growing along our elevational gradient were generally higher (i.e., less drought tolerant) than those reported from other functional groups and habitats. Specifically, the TLP of cloud forest vascular epiphytes (−2.71 to −1.67 MPa) [41], understory shrubs from a tropical rainforest (−2.55 to −1.43 MPa [73]), cloud forest trees (−2.04 to −1.34 MPa [74]), and trees from tropical rainforests (−2.18 to −1.02 MPa [75]) are all lower than what we measured in understory saplings and shrubs along our tropical montane gradient.

Other studies have also found the TLP to be related to precipitation and the VPD such that plants generally have lower TLPs (indicating a greater drought tolerance) in drier places [63,76]. The relatively high TLP and the absence of a relationship with either elevation or the MAP along our gradient (Table 4; Figure 4) may indicate that our study area was wetter than those of previous studies either due to higher rainfall, frequent cloud immersion, or higher soil water retention. In addition, understories are generally cooler and more humid than canopies [77], which may explain the lower TLP values for the trees and epiphytes. Alternatively, the plant communities that we studied may be relying more heavily on other mechanisms to tolerate periods of low water availability.

Similarly, we found no relationship of the cuticular conductance (G_min_) with either precipitation or elevation along our study gradient (Figure 4). The lowest values of the G_min_ were in the high-elevation gallery forests, and the highest values of the G_min_ occurred in the cloud forest plots at around 2500 m asl (Table 4). Cloud forests are ever-wet habitats with a high rainfall and near-constant fog immersion—conditions that are unlikely to favor drought tolerance [78,79]. A high cuticular conductance may also facilitate foliar water uptake in cloud forests [80,81,82,83,84]. If this is the case, these plants may be particularly susceptible to changes in water availability due to altered cloud immersion patterns [84].

Cell membranes are one of the first victims of many plant stresses and it is generally accepted that the maintenance of their integrity and stability under water stress conditions is a major component of the resistance to environmental stresses in plants [65]. As such, measurements of the solute leakage (SL) under standardized extreme desiccating conditions should provide valuable information about plant strategies for dealing with low water availability and stress. Indeed, SL has been previously linked to plant tolerances to salt and drought stress [58,85], high temperatures [86], and freezing [87]. In our study, we found high CWMs of the SL in plots in wetter areas and at lower elevations, while understory plant communities located in areas with low precipitation and a low temperature showed better average membrane stability (low CWMs of SL) (Table 4; Figure 4).

High-elevation shrubs had the lowest SL values (i.e., the highest drought tolerance) in the gradient, possibly due to the combination of the dry and cold (sometimes freezing) conditions that can occur in these habitats (Table 4). Lowland and high-elevation gallery forests (400 and 3548–3615 m asl, respectively) had intermediate SL values. Gallery forests are located between or below mountain ridges and may receive less fog because of topographical configuration, resulting in relatively dry conditions and potentially explaining their high drought tolerance relative to other habitats. Lowland forests experience less rainfall and thus exhibit values of SL that are comparable with gallery forests >3000 m higher on the gradient. The highest SL values were exhibited by sub-montane forests at around 1000 m asl (Table 4). The plots in these sub-montane forests receive the highest precipitation (5362–5992 mm yr^−1^) and can be extremely humid. The high SL values of the understory plants in the sub-montane forests suggest that these communities are adapted to super wet conditions and may therefore suffer due to any future climate-change-driven decreases in rainfall or humidity.

### 4.3. Solute Leakage (SL) Is a Potentially Efficient and Effective Tool for Measuring Plant Drought Tolerances in the Field

It is particularly noteworthy that the solute leakage had the only significant relationships with the environmental variables (Figure 4). The SL has previously been used in laboratory and agricultural assays, but only a few very recent studies have used it to assess the drought tolerance of plants in the field. This is surprising, given that SL has many advantages that make it particularly well suited for studying the patterns of plant drought tolerances in biodiverse communities and in remote settings [59,60]. Specifically, the protocol is easy to follow and standardize, large numbers of treatments can be performed relatively quickly, the equipment needed for the experiments have a low-cost relative to other physiological measures of drought tolerance, and the measurements can be made in areas with little or no existing infrastructure (e.g., measuring SL does not require electricity or compressed gases and uses only distilled water and polyethylene glycol). Based on the results presented here, along with the results of [59,60], it appears that differences in the water availability lead to predictable changes in the SL, indicating that SL may be a powerful tool for assessing plant–water relations. We strongly encourage future studies to help test and develop this method.

## 5. Conclusions

Studies of tropical montane forests can be particularly challenging given the large number of species involved, the frequent absence of local taxonomic treatments, the diverse array of plant habitats, and the difficulty of accessing the field sites. As a result of these and other challenges, the relationships between the water availability, elevation, and functional traits in Andean forests remain largely unknown. This study helps to advance our understanding of how the variation in water availability drives trait adaptation and community assembly in hyperdiverse tropical understory plant communities. We show that because of habitat filtering and/or local adaptation and acclimation, the plant assemblages in areas with different water availabilities have significant differences in their drought tolerance as measured by the solute leakage (but not by the G_min_ or TLP). Temperatures in the Andes are increasing rapidly, and there is an overall tendency of increasing seasonality and decreasing annual rainfall through time [32,88]. Increasing temperatures and decreasing rainfall will reduce the water availability and increase the occurrence of droughts. In addition, there is a trend of decreasing mean annual and mean monthly cloud frequencies, which could exacerbate water stress and extend the duration of the dry season [89]. Changes in the water availability along this elevational gradient will expose the plant communities to different moisture regimes and can potentially lead to changes in species’ distributions and the composition of understory plant communities [72]. Given the importance of the understory in supporting biodiversity [90], and the fact that saplings and juveniles of all canopy trees must pass through the understory, changes in the understory conditions and composition will have significant consequences for many plant and animal species as well as the valuable ecosystem services that these forests provide.

## Figures and Tables

**Figure 1 plants-12-02957-f001:**
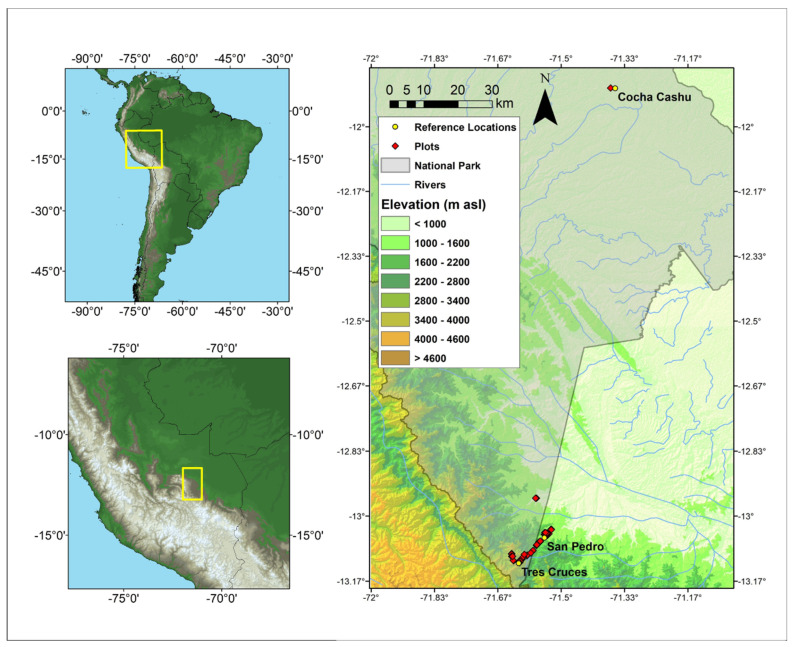
Map of the study plots along an elevational gradient in and around Manu National Park in southeastern Peru (Cusco and Madre de Dios regions). The study area ranges from 400 to 3600 m asl and includes forests along the Trocha Union ridge in the Kosñipata Valley (900 to 3600 m asl) with field stations at Tres Cruces and San Pedro, and near the Cocha Cashu Biological Station (400 m asl; Table 1).

**Figure 2 plants-12-02957-f002:**
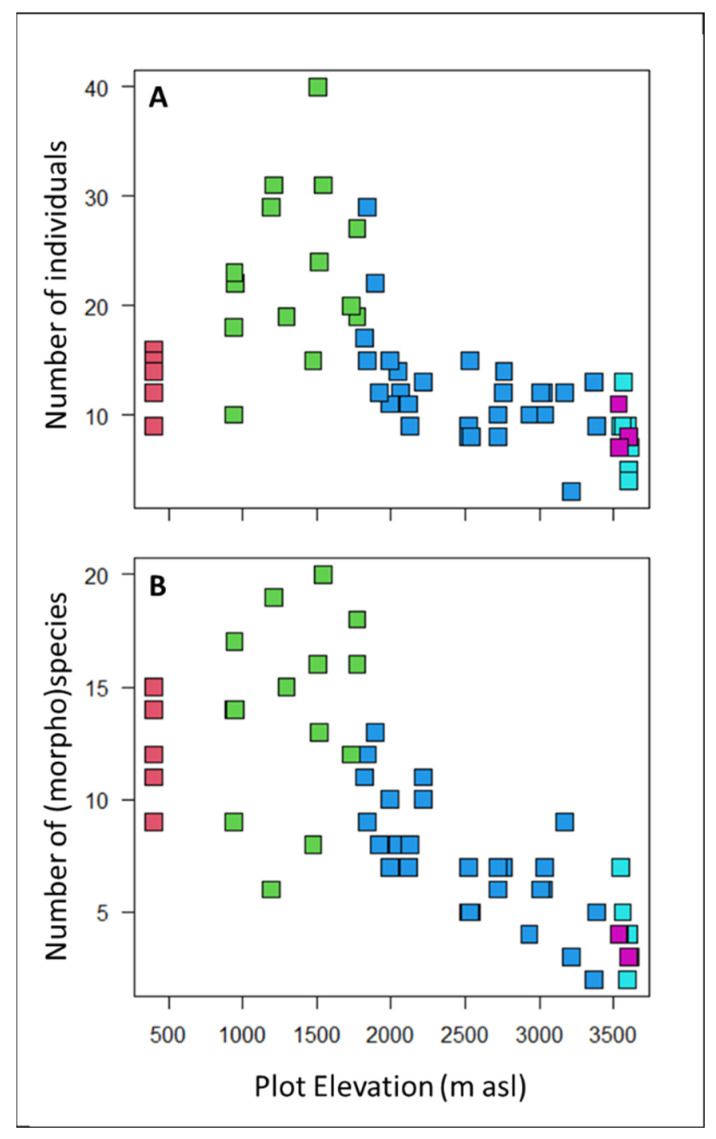
The (**A**) number of individual understory saplings and shrubs and (**B**) the number of understory species and morphospecies recorded in the 5 × 5 m study plots were greatest in sub-montane forests around 1500 m asl and then decreased steadily with elevation. Points are colored according to plot habitat as follows: red = lowland forest, green = sub-montane forest, blue = cloud forest, cyan = high-elevation gallery forest, and purple = high-elevation shrubs.

**Figure 3 plants-12-02957-f003:**
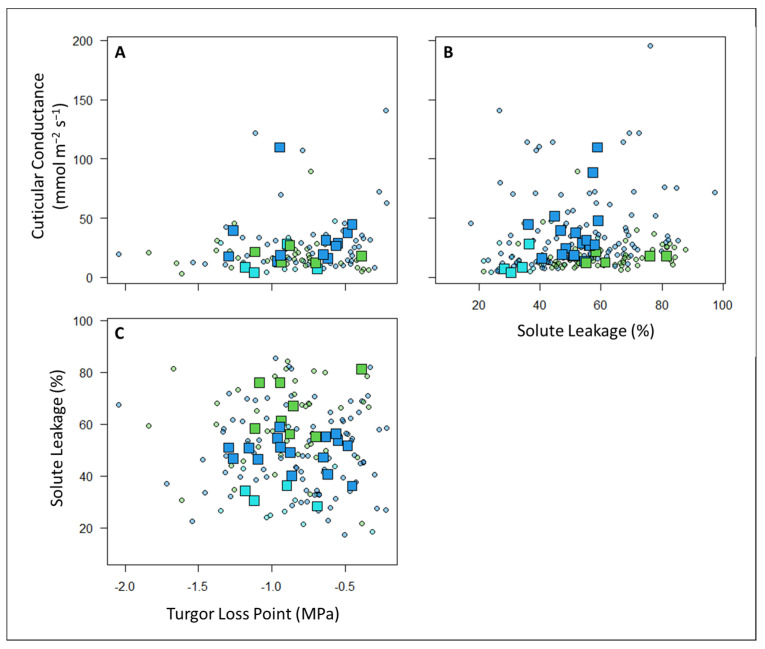
There were no significant correlations at either the individual level (circles = individual plant measurements) or the plot level (squares = community-weighted means) between (**A**) the G_min_ and TLP, (**B**) the G_min_ and SL, or (**C**) the SL and TLP. Points are colored according to plot habitat as follows: green = sub-montane forest, blue = cloud forest, and cyan = high-elevation gallery forest. Pearson correlation coefficients are listed in Table 3.

**Figure 4 plants-12-02957-f004:**
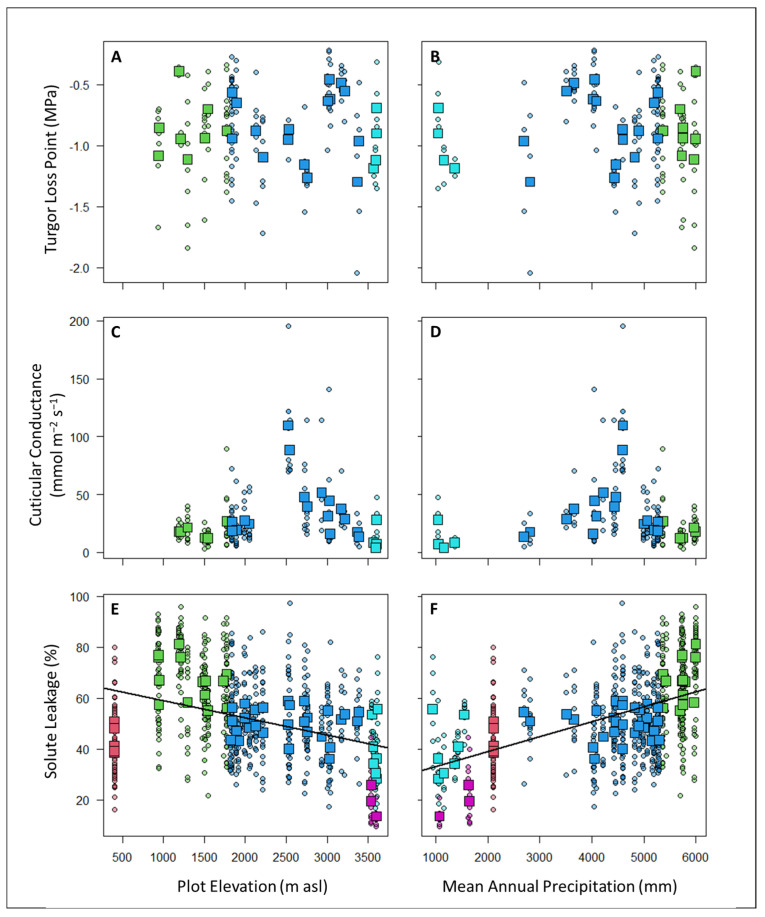
There were no significant correlations at either the individual level (circles = individual plant measurements) or the plot level (squares = community-weighted means) between the TLP (panels (**A**,**B**)) or G_min_ (panels (**C**,**D**)) with either elevation (panels (**A**,**C**)) or mean annual precipitation (panels (**B**,**D**)). In contrast, solute leakage (SL) was significantly related to elevation (panel (**E**); SL ~ 65.94–0.0067 × Elevation; *p* < 0.00005; adjusted R^2^ = 0.23) and MAP (panel (**F**); SL ~ 27.23 + 0.0058 × MAP; *p* < 0.00005; adjusted R^2^ = 0.50). Points are colored according to plot habitat as follows: red = lowland forest, green = sub-montane forest, blue = cloud forest, cyan = high-elevation gallery forest, and purple = high-elevation shrubs.

**Table 1 plants-12-02957-t001:** Locations and characteristics of study plots.

Plot	Latitude (°)	Longitude (°)	Elevation (m asl)	MAT (°C)	MAP (mm/Year)	Habitat	No. of Individuals	No. of Shrubs	No. of Saplings	No. of Families	No. of Genera	No. of (Morpho) Species
1	−13.096	−71.630	3604	8.2	1617	gallery forest	5	3	2	3	4	4
2	−13.097	−71.630	3569	8.2	1437	gallery forest	13	5	8	3	3	4
3	−13.103	−71.630	3603	8.0	1054	gallery forest	4	3	1	2	2	4
4	−13.103	−71.629	3592	18.9	1157	gallery forest	9	0	9	1	2	2
5	−13.114	−71.625	3548	18.9	1549	gallery forest	9	5	4	2	1	7
6	−13.113	−71.626	3562	10.1	1515	gallery forest	9	3	6	3	2	5
7	−13.103	−71.628	3615	12.5	937	gallery forest	7	1	6	3	2	3
8	−13.110	−71.603	3172	18.7	3654	cloud forest	12	7	5	4	5	9
9	−13.110	−71.604	3214	18.7	3507	cloud forest	3	3	0	2	3	3
10	−13.114	−71.607	3366	17.7	2812	cloud forest	13	9	4	2	2	2
11	−13.112	−71.607	3387	17.7	2692	cloud forest	9	1	8	4	4	5
12	−13.033	−71.526	1189	10.1	5992	sub-montane forest	29	21	8	5	5	6
13	−13.035	−71.526	1212	17.7	5990	sub-montane forest	31	16	15	5	5	19
14	−13.045	−71.532	1296	20.2	5989	sub-montane forest	19	7	12	10	9	15
15	−13.047	−71.544	1771	19.7	5363	sub-montane forest	27	9	18	6	8	18
16	−13.047	−71.542	1837	17.7	5266	cloud forest	29	6	23	5	6	12
17	−13.049	−71.536	1503	20.1	5744	sub-montane forest	40	8	32	6	9	16
18	−13.048	−71.537	1545	8.0	5694	sub-montane forest	31	19	12	7	8	20
19	−13.119	−71.609	3601	10.1	1082	puna shrubs	8	8	0	0	0	3
20	−13.116	−71.608	3539	12.5	1082	puna shrubs	7	7	0	0	0	4
21	−13.118	−71.611	3536	10.1	1082	puna shrubs	11	11	0	0	0	4
22	−13.101	−71.590	2754	13.3	4428	cloud forest	12	5	7	4	3	7
23	−13.101	−71.590	2762	13.6	4421	cloud forest	14	5	9	7	6	7
24	−13.100	−71.589	2721	11.2	4457	cloud forest	10	5	5	5	5	6
25	−13.100	−71.589	2724	12.5	4455	cloud forest	8	4	4	5	4	7
26	−13.089	−71.575	2525	14.4	4595	cloud forest	9	6	3	4	4	7
27	−13.089	−71.575	2526	14.4	4594	cloud forest	8	4	4	2	1	5
28	−13.088	−71.574	2542	13.6	4584	cloud forest	8	1	7	4	1	5
29	−13.094	−71.580	2532	17.7	4590	cloud forest	15	12	3	3	4	5
30	−13.104	−71.599	3036	22.1	4021	cloud forest	10	7	3	3	2	7
31	−13.104	−71.599	3025	14.5	4045	cloud forest	12	8	4	3	4	6
32	−13.104	−71.599	3006	21.8	4084	cloud forest	12	1	11	3	3	6
33	−13.098	−71.597	2933	17.7	4213	cloud forest	10	4	6	3	3	4
34	−13.070	−71.560	2064	8.7	4973	cloud forest	12	3	9	6	7	8
35	−13.070	−71.560	2048	9.2	4991	cloud forest	14	4	10	7	7	8
36	−13.068	−71.559	1994	8.2	5056	cloud forest	11	6	5	6	6	7
37	−13.067	−71.559	1992	8.2	5059	cloud forest	15	3	12	6	8	10
38	−13.065	−71.556	1891	10.1	5191	cloud forest	22	8	14	8	9	13
39	−13.066	−71.556	1920	8.2	5152	cloud forest	12	7	5	6	6	8
40	−13.065	−71.555	1839	13.3	5265	cloud forest	15	1	14	7	9	9
41	−13.064	−71.555	1820	14.4	5292	cloud forest	17	8	9	4	6	11
42	−13.074	−71.565	2217	17.7	4789	cloud forest	13	6	7	7	6	10
43	−13.075	−71.565	2121	17.7	4789	cloud forest	11	1	10	5	5	7
44	−13.074	−71.565	2217	18.7	4789	cloud forest	13	10	3	4	5	11
45	−13.073	−71.564	2130	18.7	4789	cloud forest	9	3	6	7	7	8
46	−13.044	−71.536	1474	17.7	5754	sub-montane forest	15	2	13	5	6	8
47	−13.043	−71.537	1514	17.7	5754	sub-montane forest	24	8	16	10	10	13
48	−13.042	−71.543	1772	18.9	5395	sub-montane forest	19	11	8	13	12	16
49	−13.042	−71.541	1732	18.9	5395	sub-montane forest	20	2	18	7	11	12
50	−12.954	−71.565	937	17.7	5849	sub-montane	18	4	14	9	10	14
51	−12.954	−71.566	938	18.9	5849	sub-montane forest	10	6	4	6	6	9
52	−12.954	−71.567	946	22.1	5849	sub-montane forest	22	13	9	7	8	14
53	−12.954	−71.567	944	21.8	5849	sub-montane forest	23	13	10	8	11	17
54	−11.900	−71.370	400	25.0	1366	lowland forest	9	0	9	0	0	9
55	−11.900	−71.370	400	25.0	1366	lowland forest	16	0	16	11	11	11
56	−11.900	−71.370	400	25.0	1366	lowland forest	15	0	15	0	0	15
57	−11.900	−71.370	400	25.0	1366	lowland forest	14	0	14	0	0	14
58	−11.900	−71.370	400	25.0	1366	lowland forest	12	0	12	11	12	12

**Table 2 plants-12-02957-t002:** Community-weighted means of leaf drought tolerance metrics (se = standard error).

Plot	SL (%)	SL_se	TLP (MPa)	TLP_se	G_min_ (mmol m^−2^ s^−1^)	G_min__se
1	36.40	6.74	−0.95	0.15	26.43	5.00
2	34.26	2.30	−1.20	0.02	8.87	2.04
3	28.37	4.07	−0.69	0.12	7.06	0.22
4	30.47	3.77	−1.04	0.04	4.18	0.00
5	53.59	1.90	NA	NA	NA	NA
6	41.06	1.28	NA	NA	NA	NA
7	55.77	6.78	NA	NA	NA	NA
8	51.76	4.23	−0.44	0.04	33.88	6.53
9	53.79	20.95	−0.55	0.10	28.86	3.21
10	50.96	2.07	−1.42	0.31	21.11	4.92
11	54.59	4.64	−0.96	0.22	12.73	1.65
12	81.36	1.17	−0.36	0.01	23.20	2.61
13	76.05	2.06	−0.94	0.03	22.48	1.99
14	58.32	4.62	−0.96	0.15	26.02	3.02
15	56.19	3.21	−0.93	0.09	20.46	4.26
16	56.33	2.65	−0.51	0.03	17.98	3.37
17	61.42	2.02	−0.89	0.06	11.60	0.84
18	55.12	2.16	−0.65	0.07	11.97	1.10
19	13.62	1.24	NA	NA	NA	NA
20	26.02	4.38	NA	NA	NA	NA
21	19.50	2.62	NA	NA	NA	NA
22	46.82	2.68	−1.28	0.02	35.85	7.35
23	52.47	3.55	NA	NA	NA	NA
24	50.92	5.19	−1.16	0.07	NA	NA
25	59.04	5.72	NA	NA	47.17	8.20
26	58.89	6.56	−0.96	0.04	99.11	14.28
27	49.76	6.76	NA	NA	NA	NA
28	57.37	8.18	NA	NA	96.76	8.34
29	40.11	2.95	−0.87	0.00	NA	NA
30	40.69	5.38	−0.62	0.06	15.63	2.25
31	36.21	3.47	−0.37	0.07	74.28	22.50
32	55.23	3.52	−0.60	0.06	31.21	0.00
33	44.89	2.93	NA	NA	57.21	19.77
34	50.80	4.98	NA	NA	NA	NA
35	48.40	2.65	NA	NA	21.30	5.44
36	52.36	3.51	NA	NA	NA	NA
37	57.97	3.21	NA	NA	23.95	3.87
38	47.25	2.53	−0.70	0.06	17.71	3.38
39	43.25	2.94	NA	NA	NA	NA
40	51.18	4.49	−0.96	0.09	17.76	2.39
41	43.68	2.71	NA	NA	NA	NA
42	46.59	4.25	−1.02	0.09	NA	NA
43	55.56	3.60	NA	NA	NA	NA
44	56.27	4.06	NA	NA	NA	NA
45	49.21	4.06	−0.81	0.11	NA	NA
46	66.56	3.82	NA	NA	NA	NA
47	66.67	3.39	NA	NA	NA	NA
48	69.45	3.56	NA	NA	NA	NA
49	66.79	4.22	NA	NA	NA	NA
50	75.95	3.47	−1.01	0.11	NA	NA
51	76.98	3.78	NA	NA	NA	NA
52	66.99	4.14	−0.84	0.03	NA	NA
53	57.47	3.80	NA	NA	NA	NA
54	46.46	5.40	NA	NA	NA	NA
55	38.61	3.47	NA	NA	NA	NA
56	48.30	3.35	NA	NA	NA	NA
57	41.27	3.77	NA	NA	NA	NA
58	39.19	3.52	NA	NA	NA	NA

**Table 3 plants-12-02957-t003:** Pearson correlation coefficients between the three drought tolerance metrics measured at the individual plant level (shaded values in the lower triangle) and plot level with CWMs (unshaded values in upper triangle). None of the correlations were significant (the correlation between the SL and G_min_ at the individual level was marginally significant with *p* = 0.06).

	TLP	G_min_	SL
TLP		0.067	0.092
G_min_	0.130		0.174
SL	−0.043	0.139	

**Table 4 plants-12-02957-t004:** Means (with standard deviations) of drought tolerance metrics per habitat.

Habitat	N. Plots	TLP	G_min_	SL
Lowland Forest	4	NA	NA	42.77 (4.37)
Sub-Montane Forest	5	−0.82 (0.22)	19.29 (6.08)	66.81 (8.45)
Cloud Forest	29	−0.83 (0.31)	38.38 (27.64)	50.43 (5.93)
High-Elevation Gallery Forest	7	−0.97 (0.21)	11.64 (10.05)	39.99 (10.85)
High-Elevation Shrubs	3	NA	NA	19.71 (6.21)

## Data Availability

Data are available in the article’s Appendix A.

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
