# Peer review of "Variation in the Drought Tolerance of Tropical Understory Plant Communities across an Extreme Elevation and Precipitation Gradient"

_plants, 2023, doi:10.3390/plants12162957_

Round 1

Reviewer 1 Report

Detailed comments are on the margins of the manuscript in the attached file. 

Excellent quality, slight improvement is needed to make the paragraphs more concise. 

Author Response

We appreciate Reviewer #1’s comments and suggestions. We have fully revised the manuscript including many of the changes suggested in the pdf; we are confident that the manuscript is significantly improved as a result of these changes.

Reviewer 2 Report

This research is somewhat interesting in terms of the approach that solute leakage can be an efficient ecophysiological tool for measuring drought tolerance in rainforests. However, I feel this study seems like a preliminary study rather than sufficiently conducted. Especially introduction is required to revise by focusing on the rationale of the study and current research trends regarding the topic authors conducted. In addition, the discussion part should be fully rewritten by sub-section (i.e., by your research questions and/or hypothesis; by independent variables in the methodology part) by adding more sufficient references and deeper discussion.

Even if solute leakage was related to elevation and mean annual rainfall, I still feel there are no significant findings and a lack of supporting data and evidence to conclude it for publication. 

I do not see any substantial language problem, but writing concisely might be required. 

Author Response

We appreciate Reviewer #2’s detailed critique of our manuscript. We have made many of the changes that they suggested in the pdf.  We respectably disagree with Reviewer #2’s contention that that the “study seems like a preliminary study rather than sufficiently conducted”.  In the study, which was conducted in one of the most remote and diverse field locations on Earth, we measured 1-3 metrics of physiological drought tolerance on 834 understory plants representing 363 species (89 genera and 50 families) in 58 plots distributed across a >3000 meters elevational and 5000 mm rainfall gradient. This study was conducted as part of C. Bravo A.’s doctoral dissertation and represents a huge undertaking - perhaps one of the largest studies of its kind for tropical montane forests.  We also disagree that there are “no significant findings and a lack of supporting data”.  We show that there are strong (R2= 0.5) and significant relationships between SL and elevation and rainfall.  This finding indicates that the plants are responding to differences in water availability and thus supports the hypothesis that changes in water availability associated with global warming are likely to have important effects on tropical forest community species composition.  Also important is the fact that neither TLP and Gmin are able to detect the species’ responses to annual rainfall or elevation despite the fact that these are both widely used metrics of drought tolerance. In terms of supporting data, we include a full appendix that has detailed raw data and information for every individual included in this study. That said, we do agree that more research is called for to understand the responses of tropical montane understory plants to drought and temperature differences (and we indicate as much in our revised manuscript). We have elected to not add subheadings to the discussion section as it is not excessively long, and we fear that subheadings would unnecessarily break the flow.

Reviewer 3 Report

This reviewer has only a few not very important comments. If  necessary the complete manuscript will be sent separatly to the editor in form of a separate PDF.

Please check the references again.

Author Response

We thank Reviewer #3 for their positive assessment of our manuscript.

Round 2

Reviewer 2 Report

I appreciate the authors provided a revised manuscript by providing some responses. Even if it was a big endeavor in terms of the fact that it was part of an author's doctoral thesis, in respect of results, I still do not see if there are any significant findings 

(e.g., what if using solute leakage is an efficient tool in ecology study of rainforests and this is a pioneer study regarding this?).  

and not much sufficient evidence and supporting references in the authors statement: 1) "Our results illustrate that differences in water availability may play a strong role in structuring plant communities even in the wet tropics and, supporting the hypothesis that changes in water availability associated with global warming are likely to have important effects on species distributions and community composition." and 2) "There were no significant correlations between the CWMs of SL, Gmin, and TLP". 

In addition, it was not reflected that this manuscript should be discussed each result in the sub-section and in the discussion part (L9-13). Plus, it is likely to be important to put effort into the discussion part. For instance, the authors mentioned many things without citation such as "Cloud forests are ever-wet habitats with high rainfall and near-constant fog immersion – conditions that may not select for traits conferring drought tolerance. High cuticular conductance may also facilitate foliar water uptake in the cloud forests.", which means it is still needed deeper discussion with more references.

Lastly, some format is not ready for publication such as references format in sentences, line numbers, and irregular use/omitting of abbreviation. For these reasons, I do not feel this manuscript is not ready to publish yet.

I just feel it is too late to change the whole thing at this moment.

Thank you,

Author Response

I appreciate the authors provided a revised manuscript by providing some responses. Even if it was a big endeavor in terms of the fact that it was part of an author's doctoral thesis, in respect of results, I still do not see if there are any significant findings (e.g., what if using solute leakage is an efficient tool in ecology study of rainforests and this is a pioneer study regarding this?).  and not much sufficient evidence and supporting references in the authors statement: 1) "Our results illustrate that differences in water availability may play a strong role in structuring plant communities even in the wet tropics and, supporting the hypothesis that changes in water availability associated with global warming are likely to have important effects on species distributions and community composition." and 2) "There were no significant correlations between the CWMs of SL, Gmin, and TLP". 

RESPONSE: We are grateful for the opportunity to respond to the reviewers’ comments and to revise our manuscript to address their concerns.  We appreciate the reviewer’s suggestions - we have made many revisions throughout the manuscript ad it is significantly improved as a result.

We are confident that our results are significant and important.  As the reviewer points out “solute leakage is an efficient tool in ecology study of rainforests and this is a pioneer study regarding this”. As such, we expect that this paper can help to facilitate exciting new research into the drought tolerance of plants, especially in diverse and remote systems which are too often understudied due to logistical challenges. In addition, we provide valuable data about the physiology and ecology for a large number of species in an especially hyperdiverse, understudied and severely threatened habitat.  These results - and the associated datasets - help to increase our understanding of how tropical understory plants adapt to their environment.

In accord with the reviewer’s specific concerns, we have toned down the conclusion statement in our Abstract so that we no longer state that our results can be used to help predict responses to climate change. Regarding the “correlations between the CWMs of SL, Gmin, and TLP” we have revised the associated text in the manuscript, and we have added a new table (Table 3) showing the Pearson correlation coefficients between each drought tolerance mechanism at both the individual and plot levels.  These relationships are also graphically illustrated in figure 3.

In addition, it was not reflected that this manuscript should be discussed each result in the sub-section and in the discussion part (L9-13). Plus, it is likely to be important to put effort into the discussion part. For instance, the authors mentioned many things without citation such as "Cloud forests are ever-wet habitats with high rainfall and near-constant fog immersion – conditions that may not select for traits conferring drought tolerance. High cuticular conductance may also facilitate foliar water uptake in the cloud forests.", which means it is still needed deeper discussion with more references.

RESPONSE: We have fully revised the manuscript's Discussion section. As per the reviewer’s suggestion, we have now broken the Discussion section into subsections - we thank for the reviewer for this suggestion as we agree that the Discussion is much clearer now. We have added many more references to the text,  including several citations related to the cloud forest hydrological conditions and foliar uptake.  We have added additional text discussing the importance of foliar uptake.

Lastly, some format is not ready for publication such as references format in sentences, line numbers, and irregular use/omitting of abbreviation. For these reasons, I do not feel this manuscript is not ready to publish yet.

RESPONSE: We look forward to working with the journal’s copy editors to ensure that our manuscript is formatted correctly for publication in Plants.

I just feel it is too late to change the whole thing at this moment.

RESPONSE: We appreciate the editor allowing us to revise and resubmit our manuscript.